# Biochemical Analysis of Wheat Milling By-Products for Their Valorization as Potential Food Ingredients

**DOI:** 10.3390/ijms26125830

**Published:** 2025-06-18

**Authors:** Chiara Suanno, Lorenzo Marincich, Simona Corneti, Iris Aloisi, Luca Pincigher, Elisa Papi, Luigi Parrotta, Fabiana Antognoni, Stefano Del Duca

**Affiliations:** 1Department of Biological, Geological and Environmental Sciences, University of Bologna, Bologna, Via Irnerio 42, 40126 Bologna, Italy; chiara.suanno3@unibo.it (C.S.); simona.corneti2@unibo.it (S.C.); iris.aloisi2@unibo.it (I.A.); stefano.delduca@unibo.it (S.D.D.); 2Department for Life Quality Studies, University of Bologna, Rimini Campus, Corso d’Augusto 237, 47921 Rimini, Italy; lorenzo.marincich2@unibo.it (L.M.); fabiana.antognoni@unibo.it (F.A.); 3Department of Pharmacy and Biotechnology, University of Bologna, Bologna, Via Belmeloro 6, 40126 Bologna, Italy; luca.pincigher2@unibo.it; 4Molino Pivetti S.P.A., Via Renazzo, 67, 44045 Renazzo, Italy; elisa.papi@pivetti.it; 5Interdepartmental Centre for Agri-Food Industrial Research, University of Bologna, Bologna, Via Quinto Bucci 336, 47521 Cesena, Italy

**Keywords:** biowaste upcycling, food security, nutraceutical potential, human nutrition, bakery products, food sustainability, *Triticum aestivum*, circular economy

## Abstract

Wheat bran forms the outermost part of the kernel, which is typically discarded as a by-product. Depending on the milling process, bran can be separated into four fractions: coarse bran (CB), coarse weatings (CW), fine weatings (FW), and low-grade flour (LGF). This study aimed to analyze the macronutrient and bioactive compound profiles of these four by-products across five cultivars and two wheat mixtures. Dietary fibers, free and bound phenolics, phytic acid, fatty acids, and aleurone layer markers were examined in all samples. The results indicate that insoluble fibers, phenolic compounds, and phytic acid decreased from CB to LGF, whereas soluble fiber content exhibited a greater variability among fractions. In all samples, coarse bran was the richest fraction in the protein 7S globulin. The same fraction from the two commercial mixtures and Manitoba cultivar exhibited significantly higher levels of bound ferulic acid compared to the other cultivars (+34%). Manitoba CB also had the highest oleic acid content (18.04% of total lipid content) among all samples, followed by the Rumeno cultivar (17.75%), which also had the highest linolenic acid content (6.35%). Given their health-promoting and technological potential, these by-products could be selectively used to enrich food products and dietary supplements with functional nutrients.

## 1. Introduction

While food security is a priority recognized by various international agencies [1,2], food production is closely linked to the over exploitation of natural resources. It is estimated that 17% of food produced is lost or wasted between harvest and retail [3]. To address this issue, one of the key targets of the United Nations’ 2030 Agenda for Sustainable Development [4] is to significantly reduce waste generation through prevention, reduction, recycling, and reuse. The Circular Economy Action Plan emphasizes that achieving this goal requires a shift from a linear economy, where products are destined to become non-reusable waste, to a circular economy, in which the end of one product’s life cycle initiates another industrial process [3].

The agri-food industry generates large amounts of waste and by-products, which are often rich in valuable compounds that can be repurposed in nutraceutical, food, cosmetic, pharmaceutical, and packaging applications, thus promoting a circular economy and reducing environmental impact [5,6,7,8,9,10,11,12]. A notable example is the common wheat (*Triticum aestivum* L.), one of the most widely cultivated food crops and a staple in many countries worldwide. During wheat grain processing, the starchy endosperm, comprising 80–85% of the dry weight, is primarily used for food production, while the remaining components, including the germ and the multiple outer layers, are removed during milling. These layers include the aleurone layer (the outermost part of the endosperm), the hyaline layer and testa (forming the seed coat), and the pericarp [13,14,15].

In the conventional roller milling process, the endosperm is separated from the germ and bran through successive grinding, sieving, and purifying steps. The endosperm is then further refined into flour, which is traditionally used for baking. The outer layers, still containing remnants of starchy endosperm and germ, are recovered at different milling stages, forming milling by-products (MBPs), commonly known as bran [16]. Wheat milling generates approximately 23–27% MBPs, with an annual global production exceeding 150 million tons [17]. Of this, 90% is used in livestock feed, while only 10% is allocated for human consumption [18,19]. MBPs represent an important source of dietary fiber, consisting of a complex blend of cellulose, hemicellulose, and pentosan polymers linked to proteins [20]. This indigestible fraction of plant cell wall polysaccharides resists enzymatic digestion in the upper gastrointestinal tract, reaching the colon, where it undergoes bacterial fermentation [21]. Grains contain 33.4–63.0% of their dietary fiber in the bran, which accounts for many of the recognized health benefits associated with whole grain consumption [13]. Dietary fibers are classified as either ‘soluble’ or ‘insoluble’ based on their water solubility [19].

From a nutritional perspective, MBPs are composed of starch and non-starch polysaccharides (including dietary fiber), lignin, proteins, lipids, and minerals [22]. Among these components, the aleurone layer is of particular scientific and nutritional interest due to its high concentration of bioactive compounds. It is especially rich in β-glucans, proteins (mainly globulins), highly substituted arabinoxylans, esterified ferulic acid monomers, and essential minerals such as Fe, Mg, Zn, and Ca, as well as nearly all B-group vitamins [23,24]. These characteristics make MBPs, and specifically the aleurone layer, a promising source of micronutrients with significant potential in human nutrition [25]. However, several structural and biochemical factors limit its full exploitation. Aleurone cells, which store valuable proteins and nutrients, are encased in a matrix of insoluble complex carbohydrates and lignin, which hampers their digestibility in the human gastrointestinal tract [25]. In addition, MBPs contain phytic acid, a natural antioxidant that strongly binds to storage proteins and minerals, reducing their solubility and bioavailability [24]. Phytic acid constitutes approximately 1–5% of MBP weight and accounts for 60–90% of the total phosphorus content in cereal seeds used for food and feed [26].

MBPs vary depending on the milling stage at which they are produced and can be classified into four categories: coarse bran (CB), also known as regular bran; coarse weatings (CW), also called fine bran; fine weatings (FW), also known as middlings or shorts; and low-grade flour (LGF), also referred to as red dog [16]. CB consists of all outer layers along with traces of starchy endosperm. It has a wide particle size distribution (~1153 µm) and is composed of approximately 51% dietary fiber, 22% starch, 18.9% proteins, 5.4% ash, and 2.8% lipids. CW is obtained later in the milling process and has finer particles (~449 µm) with a relatively higher proportion of endosperm. Its composition includes 43.2% dietary fiber, 28.8% starch, 18.8% proteins, 4.6% ash, and 3.7% lipids. FW and LGF have even finer particle sizes (~249 µm for FW, ~173 µm for LGF); the former contains 31.9% dietary fiber, 40.5% starch, 19.5% proteins, 3.8% ash, and 5.0% lipids, while the latter has 14.5% dietary fiber, 54.4% starch, 18.8% proteins, 2.8% ash, and 5.7% lipids [15,16] (Figure 1).

Although 90% of reused MBPs are destined for animal feed production, the number of food products enriched with these by-products is steadily increasing. MBPs are widely used in the “bakery and cereals” sector, which includes breakfast cereals, bread, cereal bars, and rolls [17]. However, the incorporation of these by-products into cereal-based foods remains a challenge because they alter the organoleptic properties, often resulting in less appealing final products to consumers [15,27,28]. Although each MBP has the potential to improve dietary fiber levels in bread [16,29,30], only a limited number of studies have explored the functionality of each MBP in bread making, highlighting how they differently affect the final product [16,29,30]. Moreover, the characterization of each MBP from different commercial mixtures is still overlooked, as the common approach is to counterbalance their negative effects by using strong wheat flours and adding bread improvers [16,29,30]. Therefore, describing of the features of different MBPs from different commercial mixtures is critical, as selecting them based on their specific properties to match specific formulations could improve product formulation and enhance the efficient use of by-products within a circular economy framework [15,27,31].

This study investigates the nutritional and nutraceutical composition of CB, CW, FW, and LGF derived from five commercial wheat cultivar (C1–C5) and two commercial wheat mixtures (M1–M2), aiming at analyzing their features and providing a tool for a selective choice of the best by-product to be included into a functional food. The nutritional composition was analyzed in terms of soluble and insoluble fiber, starch, α-amylase enzyme activity, fatty acids, phytic acid, and 7S globulin levels, while the nutraceutical one was evaluated through the identification and quantification of soluble-conjugated and insoluble-bound phenolic acids through HPLC-DAD.

## 2. Results

### 2.1. Insoluble and Soluble Fibers in MBPs

In all MBP samples, soluble and insoluble fibers were quantified, and their distribution among cultivar and mixtures and MBPs is illustrated in Figure 2A,B. Soluble fiber concentrations ranged from 1.80 to 3.83 g/100 g (Figure 2A), whereas insoluble fibers varied from 11.6 to 42.6 g/100 g (Figure 2B), thus representing the major fraction.

Regarding soluble fibers (Figure 2A), a significant variability was observed among the MBPs across most commercial cultivar and mixtures. The only exceptions were M1 and M2, where all four MBPs showed similar concentrations. These variations make it difficult to identify which MBP was the richest in soluble fibers, and, as a general trend, all cultivar and mixtures exhibited comparable levels. Specifically, CB, CW, and LGF did not differ significantly among cultivar and mixtures, while FW was notably higher in cultivar C4 compared to C1 and C2.

In contrast, for insoluble fibers (Figure 2B), the trend turned clearer, with CB and CW consistently emerging as the richest MBPs across all cultivar and mixtures, with no significant differences among mixtures and cultivar, except in M2 and C5, where CB contained significantly more insoluble fibers than CW. The MBPs with the lowest insoluble fiber content were FW and LGF, with LGF in C4 being significantly lower than the other MBPs.

### 2.2. Soluble-Conjugated and Insoluble-Bound Phenolic Acids in MBPs

*p*-Coumaric and ferulic acids were detected in all extracts prepared from the four MBPs derived from each commercial cultivar and mixture, in both their free (F) form and in the combined soluble-conjugated and insoluble-bound (SC+IB) fraction. Calibration curves were established for quantification of these phenolic acids, yielding the following linear regression equations: y = 120566x + 2363.2 for *p*-coumaric acid and y = 117088x + 59331 for ferulic acid.

All MBPs from the different wheat cultivars and mixtures contained ferulic acid in both the F and SC+IB forms, whereas *p*-coumaric acid was only detectable in the SC+IB fraction. The free form of ferulic acid was two to five orders of magnitude lower than the SC+IB form, with concentrations ranging from 0.005 to 0.025 mg/g DW.

The SC+IB fraction of *p*-coumaric acid was found in all MBPs, with concentrations ranging from 0.03 to 0.1 mg/g DW (Figure 3A). In general, CB and CW exhibited higher concentrations of *p*-coumaric acid than FW and LGF, although in commercial cultivar C3, C4, and C5 the differences were not statistically significant. Conversely, significant differences were observed when comparing the same MBP across different commercial cultivars and mixtures. Indeed, CB from M1 and M2 showed significantly higher levels of *p*-coumaric acid than those from the pure cultivar, which had comparable concentrations. For CW, M1 displayed the highest levels, and C1 and C5 the lowest, with the other cultivar and mixtures showing similar values. Regarding FW, mixture M2 had the highest concentration of *p*-coumaric acid, while C1, C3, and C4 had the lowest. As for LGF, C1 had the lowest content, significantly lower than that of C4, M1, and M2.

The pattern of SC+IB ferulic acid (Figure 3B) mirrored that of *p*-coumaric acid, though with much higher concentrations (ranging from 0.77 to 3.53 mg/g DW). As expected, CB and CW consistently contained the highest levels in all commercial cultivars and mixtures; in C1 and M1, these levels were significantly greater than those in FW and LGF. In C5, only CB exhibited a significantly higher ferulic acid content compared to the other MBPs. When examining inter-sample differences, CB from M1, M2, and C5 had significantly higher concentrations of ferulic acid than the other ones. Additionally, mixture M1 also had the highest levels in the CW fraction, while cultivar C1 and C4 showed significantly lower levels in FW compared to the other.

### 2.3. Polyunsaturated, Monounsaturated or Saturated Fatty Acids in MBPs

Across all commercial cultivars/mixtures and MBPs, the proportions of saturated (SFA) and monounsaturated (MUFA) fatty acids relative to the total lipid fraction were similar, ranging from 15.81% to 19.2% for SFA and from 15.25% to 19.16% for MUFA (Figure 4A–C). In contrast, polyunsaturated fatty acids (PUFA) were more abundant, accounting for between 63.82% and 67.42% of the total lipids.

Among the SFAs, palmitic acid was the only fatty acid making a significant contribution to the MBP composition (Figure 4A). Its concentration did not vary significantly among MBPs within the same cultivar and mixture, except in C3, where FW exhibited a significantly higher level than CB. However, when comparing cultivars/mixtures, the overall high palmitic acid levels in C1 were significantly different from those in C4 for CB, CW, and FW, but not for LGF.

Oleic acid was the most abundant in MUFA (Figure 4B). Generally, no significant differences in oleic acid content were observed among the four MBPs within each commercial cultivar and mixture, except for C3, where CB was notably richer in oleic acid than both CW and LGF. In the inter-cultivar/mixture comparison, C5 showed overall higher levels of oleic acid, with significant differences observed against C1, C2, C3 and C4 for the LGF fractions, against C1, C2 and C4 for FW fraction, C1 and C2 for CW fraction and C2 and C4 for CB fraction.

Finally, the predominant PUFAs were linoleic and linolenic acids. Linoleic acid did not exhibit any significant differences either within or among cultivars and mixtures, whereas linolenic acid concentrations differed significantly only between the CB fractions of C2 and C3 (Figure 4C).

### 2.4. Phytic Acid in MBPs

Phytic acid was quantified in all MBPs, with concentrations ranging from 1.02 to 3.90 g/100 g (Figure 5). In general, its concentration decreased with the increasing fineness of the MBP. In C1, C3, C5 and M2, CB contained significantly higher concentrations of phytic acid than the other MBPs. In all the cultivars and mixtures analyzed, CW was always significantly different from LGF, except in C2 and M2. Similarly, FW was consistently significantly different from CB across all samples. FW and LGF generally did not show significant differences in any of the cultivars and mixtures, except in C4, where LGF exhibited a significantly lower phytic acid concentration compared to FW.

The inter-cultivar/mixture comparison revealed that CB had the highest phytic acid content in C1, while having the lowest in M1. For CW, C3 and M2 had the lowest levels, significantly lower than those in C1 and C4. Considering the FW fraction, C4 had the highest content, significantly higher than that of C1, C3, M1, and M2; conversely, in the LGF fraction, C2 exhibited the highest content, significantly higher than that of C1, C3, C4, and M1, with C4 showing the lowest LGF content, significantly lower than that of C2, C5, and M2.

### 2.5. α-Amylase Activity and 7S Globulin in MBPs

α-Amylase activity was measured in all MBPs across the commercial cultivars and mixtures (Figure 6A). Apart from C1, significant differences in α-amylase activity among the MBPs were observed in all samples. These differences were mainly driven by the low activity in CB and, in some cases (C3, C4, and M1), by the high activity in FW. Overall, the α-amylase activity ranged from 14.21 to 45.40 CU/g. In inter-cultivar/mixture comparison, the significant differences were primarily due to the lower α-amylase activity in the CB fractions of C2 compared to those of C1, C3, and C5, and to the higher activity levels in CW and FW in C3.

The content of 7S globulin (Figure 6B) was consistently higher in CB than in the other MBPs across all commercial cultivars and mixture, with statistically significant differences in C1, C3, and M2 when compared to all other MBPs. The 7S globulin concentrations ranged from 0.007 to 1.192 Abs450/mg. In addition, significant differences were found between CW and FW in C1, and between LGF and the other MBPs (CW and CB) in M1. When comparing the same MBP across different commercial cultivars and mixtures, the coarse fractions CB and CW generally did not show significant variations in 7S globulin concentration, with the only exception of CW in C5, which differed significantly from both C4 and M1. In contrast, the FW fraction clearly divided the cultivars/mixtures into two groups: C2, C5, M1, and M2 exhibited lower 7S globulin concentrations compared to C1, C3, and C4. Considering the LGF, C3 had the highest levels of 7S globulin, whereas M1 had the lowest.

### 2.6. Starch Content in MBPs

Starch concentrations showed high variability within the commercial cultivars and mixtures, with the expected trend of increasing starch content from the coarse to the fine fractions (Figure 6C). Overall, starch content ranged from 9.6 to 50.0 g/100 g, with significant differences observed among the MBPs within each cultivar/mixture, except in C4, where CB and CW did not differ significantly, and in C5, where the only marked difference was between CB and LGF. For the CB fraction, C5 recorded the lowest starch content, significantly different from all other cultivar and mixtures except C1 (which itself had significantly lower starch content than C4, M1, and M2). For CW, the highest starch concentration was found in C5, whereas C1, C2, C3, C4, and M1 displayed significantly lower levels. In the FW fraction, C1 showed the highest starch content when compared to the FW fractions of the other cultivars/mixtures. Similarly, the LGF fraction of C3 also shows the highest starch content, which differed significantly from the other cultivars/mixtures, except for C1. In fact, the LGF fractions of C1 and C3 had the highest starch concentrations, significantly exceeding those of C2, C5, and M2.

## 3. Discussion

Dietary fibers are known to play an important role as prebiotics, since they are fermented by the microbiota, inducing the formation of short-chain fatty acids (SCFA) and the fecal bulking, thus contributing to prevent diet-related chronic diseases [29]. While cohort studies have demonstrated that insoluble cereal fiber and whole-grain products might decrease the risk of some non-communicable diseases (e.g., type 2 diabetes, cancer, and cardiovascular diseases [32,33]; intervention studies have demonstrated that the fermentation of cereal soluble fiber to SCFA can promote anti-hyperglycemic and anti-inflammatory processes on the short term [34].

Wheat milling by-products (MBPs) are known to be an important source of insoluble fibers [35]. Indeed, in our analyses, the concentration of insoluble fibers in MBPs resulted on average approximately 11 times higher than that of soluble ones (Figure 2B vs. Figure 2A), and their pattern showed a progressive decrease from the coarser to the finer by-product, confirming what it has been previously reported [16]. However, the results indicate that significant differences were found among the different cultivars and mixtures in the insoluble fiber content, and this is especially true for the intermediate and finest MBPs (FW and LGF). This demonstrates that a thorough evaluation of cultivars, e.g., Manitoba (C5) and Capo Austriaco (C2), or mixtures, e.g., Misto Ungherese (M2), would allow one to employ more refined and appealing MBPs like FW and LGF as food supplements of dietary insoluble fibers.

Although soluble fibers did not show a clear pattern across the different MBPs or commercial cultivars and mixtures (Figure 2A), this variability could represent a particularly promising opportunity for further exploration. In fact, varieties like Axioma (C1), Capo Austriaco (C2), Rumeno (C3), and Soisson (C4) could be exploited for their high variability in soluble fibers content among the different MBPs, as they offer a range of options for selecting the optimal MBP as a soluble fiber source, according to the desired technological properties.

An important factor to consider when evaluating the fiber content for possible nutritional applications is the presence of phenolic acids. In fact, most phenolic acids in wheat are covalently bound to cell wall arabinoxylans [36], thus forming the so-called “antioxidant dietary fiber” [37]. Although these phenolic compounds have been shown to have health-promoting functions, increased levels of phenolic acids bound to hemicelluloses were demonstrated to inhibit the fermentation kinetics of the latter [38], decreasing their potential as a prebiotic food constituent. Thus, a dual function for bound phenolic acids can be recognized: on one side, they are known to confer antioxidant properties to the whole grains, and on the other side they can affect the prebiotic potential of dietary fiber to which they are bound. Moreover, the interactions between fibers and phenolic acids can affect their bio-accessibility, with bound phenolics being less bio-accessible than the free ones [39]. Hence, although MBP fractions with the highest *p*-coumaric and ferulic acid levels, i.e., the CB and CW of the cultivar Manitoba (C5) and of the two mixtures Misto Rosso Baby Food (M1) and Misto Ungherese (M2) could potentially be considered as a source of antioxidant bioactive compounds (Figure 3), the presence of high levels of phenolic acids might hinder the employment of such products as prebiotic supplements. However, there is no perfect overlap between phenolic acid and insoluble fiber concentrations across all cultivars and mixtures, leaving room for a targeted selection of cultivar and fraction based on the nutrient of interest. In fact, the cultivar Soisson (C4) has an FW fraction with low phenolic acids levels and high insoluble fiber content (Figure 2 and Figure 3), so this fine fraction could be more efficiently integrated in foods with a prebiotic function.

Another natural antioxidant present in wheat is phytic acid, which is abundant in both human and animal diets [40,41]. Although phytic acid was traditionally considered an antinutritional factor due to its strong ability to chelate essential minerals such as iron, zinc, and calcium, thereby reducing their bioavailability in the gastrointestinal tract [40], more recent evidence suggests that this antinutritional effect is primarily significant in individuals with inadequate mineral intake [42]. Specifically, the inhibitory effect of phytic acid on mineral absorption becomes relevant when the dietary intake of these minerals falls below recommended levels, such as the RDAs set by health authorities (e.g., <1000 mg/day for calcium, <8–18 mg/day for iron, and <8–11 mg/day for zinc) [43,44]. On the contrary, numerous studies have demonstrated that its antioxidant potential [40,45] has preventive and therapeutic effects on various diseases, including hyperlipidemia, inflammatory bowel disease, neurodegenerative and cardiovascular diseases, kidney stone formation, and cancer development [42,45,46,47].

In the present study, the distribution of phytic acid among commercial samples and MBPs showed a general decrease going from the coarsest to the finest MBP, with CB having in most cases significantly higher levels of phytic acid than the other fractions, thus showing a similar pattern than insoluble fibers and phenolic acids. Considering the differences among commercial cultivar and mixtures, CB of cultivars like Axioma (C1) and Manitoba (C5), containing relevant amounts of phytic acid, could be used to fortify foods with phytic acid, while the CB of other cultivar or mixtures, like Misto Rosso Baby Food (M1), which has the lowest amount, should be preferred as a source of fibers in diets with low mineral intake or for subjects with deficiencies in mineral absorption.

Fatty acids are minor components of wheat grain, accounting for between 2 and 3% of its dry weight, and they are mostly concentrated in the germ [48]. Nonetheless, their presence and composition are important for both the nutraceutical and technological properties of wheat products. In fact, lipids have been demonstrated to affect volume and morphology of different wheat products (bread, steamed bread, short pastry) [49]. Moreover, when dough is prepared by adding water to wheat flour, the presence of lipoxygenase (LOX) leads to the formation of volatile compounds through the oxidative decomposition of unsaturated fatty acids. These reactions affect the flavor of processed wheat flour products, such as bread and noodles [50,51]. In addition to LOX, the flavor of wheat flour products is also influenced by unsaturated fatty acids concentration, and antioxidant factors present in the wheat cultivar [52]. Unsaturated fatty acids are the predominant type found in wheat grains. These mainly include two essential fatty acids, linoleic acid and linolenic acid, as well as oleic acid, which is a non-essential monounsaturated fatty acid. This composition was reflected by the MBPs analyzed hereby, although the concentration of each fatty acid varied according to the commercial cultivar and mixtures, and in some cases also according to the MBP, with unpredictable trends (Figure 4). In fact, while in most cases all the MBPs from the same sample showed comparable levels of the same fatty acid, Rumeno (C3) displayed some significant variations among MBPs, indicating this cultivar as possible source of different fatty acid depending on the MBP selected. This possibility is particularly appealing considering the different health effects of the three fatty acids. In fact, the presence of linoleic acid in wheat products has been suggested to correlate with beneficial effects on human cardiovascular, nervous, and immune systems, while oleic acid is thought to have a protective role against chronic illnesses [53]. It has been observed that the enrichment of flour with wheat germ improves the glycemic control in adults classified as overweight [54]. Palmitic acid, a saturated fatty acid found in wheat grains, has an ambiguous nutraceutical value. On one hand, as a saturated fat, it may contribute to increased blood cholesterol levels and negatively impact cardiovascular health, but on the other hand, it also plays a role in regulating lipid metabolism and inflammatory responses [53].

The wheat product fortification with beneficial lipids could be performed for example by adding CB from cultivar like Rumeno (C3), which retains high levels of oleic and linoleic acids, but low levels of palmitic acid, or employing more refined MBPs from other cultivar like Manitoba (C5), which has relatively low levels of palmitic and linoleic acid, but high levels of oleic acid. For example, bread fortification with fibers or lipids is a common practice, and the selection of the appropriate commercial cultivar and mixture and MBP would allow one to implement both nutrients with a reduction in the ingredients and a reuse of food waste.

The aleurone layer contains not only valuable wheat micronutrients and specific metabolites, but also a high concentration of proteins (22.9 g/100 g). These proteins mainly have a storage function, like albumins and globulins [55], but there are also enzymes, such as cellulases, proteases, lipases, oxidases, and α-amylases.

The storage proteins in the aleurone layer are considered nutritionally superior to those found in the rest of the endosperm, due to their higher content of essential and conditionally essential amino acids such as lysine, arginine, and glycine [24]. These proteins are also relevant from a technological point of view in the processing of wheat products. In fact, recent research showed that the addition of wheat globulin to dough weakens the S-S bonds in the gluten network and improves rheological properties and quality of cooked noodles [56]. Thus, the integration of proteins from the aleurone layer into wheat products is desirable, and it can be achieved by selecting fine MBPs with higher protein content, to minimize the impact on the product structure and flavor. In the present study, 7S globulin was more concentrated in CB than in the other MBPs for all commercial cultivars and mixtures, and cultivars like Soisson (C4) or mixtures like Misto Ungherese (M2) could be considered as protein sources in bran-based products. However, there were also differences in the protein content among the same MBP of different cultivars and mixtures, and FW from Axioma (C1) or LGF from Rumeno (C3), which have higher 7S globulin levels than coarser MBPs from other cultivars and mixtures, could be integrated in bakery products to improve the protein content without altering their texture.

The enzymatic activity of the MBPs is also an important aspect to consider when designing whole-grain products. In fact, enzymes play a relevant role in the baking industry [57,58,59,60] and tend to be perceived as “more natural” than food additives by the consumers [61]. In particular, α-amylases, which are naturally present in wheat, are widely employed in the baking processes since they randomly hydrolyze α-1,4 glycosidic linkages in polysaccharides, producing short chains that can be further fermented by yeasts. In baking, it is essential that the α-amylase level is adequate to hydrolyze starch to fermentable sugars, but not so high as to cause excessive starch dextrinization during the baking process, which negatively affects the quality of the product [62]. Hence, when integrating MBPs in the dough to produce whole-grain bakery products, it is important to consider the desired level of α-amylase activity for the product. In fact, if this level is already optimal, it would be preferable to select MBPs of cultivars or mixtures with low activity, like CB of Capo Austriaco (C2) or LGF of Soisson (C4) (Figure 6A). Conversely, if α-amylases need to be added to the dough, the best choice would be to select MBPs with high α-amylase activity, such as FW from Rumeno (C3).

Enzymatic activity is influenced by weather conditions and genetic differences among cultivars, as previously suggested [63], and by the availability of the substrate. In general, starch from MBPs shows a greater resistance to enzymatic digestion than that from the endosperm [64], despite having smaller granules that are thought to be more easily hydrolysable [65]. Since postprandial glycemia is directly related to the efficiency of starch digestion, the intake of resistant starch could be desirable in subjects with weight or hyperglycemia problems [66]. The assimilation of starch could be furtherly hindered by reducing the activity of starch-digestive enzymes including α-amylase and α-glucosidase, or by increasing the content of insoluble dietary fiber, which has the potential to reduce the rate of starch digestion [67].

On the other hand, insoluble fibers are the main responsible for the poor rheological quality of whole-grain baking products, compared to the flour-based ones [68]. The selection of finer MBPs with low levels of insoluble fibers and starch and low α-amylase activity might be an option to integrate MBPs nutrients in bakery products without compromising their quality and glycemic index. In fact, while CB is the fraction with the lowest levels of starch, but highest insoluble fiber content, for all the commercial cultivars and mixtures (Figure 2B and Figure 6C), the fine MBPs of some samples have significantly lower content in starch than others. This is the case for the cultivar Soisson (C4), which also shows relatively low levels of α-amylase activity but a high insoluble fiber content, or Capo Austriaco (C2), which has slightly higher levels of α-amylase activity, but a lower content in insoluble fibers than Soisson (C4).

## 4. Materials and Methods

### 4.1. Chemicals

All chemicals and reagents were bought from Sigma Aldrich (Milan, Italy) or from Merck (Darmstadt, Germany) unless differently mentioned.

### 4.2. Wheat Material

Five cultivars (C1–C5) and two commercial mixtures (M1–M2) of modern common wheat from different origins were considered for this study. The seven cultivars or mixtures used in this work were selected by the company based on commercial viability and production requirements. Grains were milled by Molino Pivetti S.P.A. (Renazzo, Ferrara, Italy) using the process follows the standard industrial wheat milling scheme. This process involves the sequential use of roller mills, plansifters, and purifiers to progressively reduce particle size, from coarse bran to low-grade flour. MBPs are collected at specific stages based on particle size. All resulting products are stored in by-product silos at ambient temperature, and samples are preserved under the same conditions. To obtain “reground” samples, an additional step is required beyond the classic milling process. Specifically, larger fractions such as coarse bran and coarse weatings undergo further regrinding using a dedicated roller mill, followed by sieving operations to separate them into final products according to particle size. The same milling operations were used to obtain CB, CW, FW, and LGF for each of the studied commercial cultivars and mixtures. Cultivars and mixtures, together with their sample code, composition and origin, are reported in Table 1.

### 4.3. Quantification of Fibers, Fatty Acids, Starch and Phytic Acid

The content of soluble and insoluble fibers, saturated and unsaturated fatty acids, starch, and phytic acid was analyzed by Mérieux NutriSciences Chelab (Resana, Treviso, Italy).

Insoluble and soluble fibers were determined using AOAC Official Method 991.43. This enzymatic-gravimetric method involves sequential digestion with thermostable α-amylase, protease, and amyloglucosidase to simulate human gastrointestinal conditions. Insoluble fibers were separated by filtration, while soluble fibers were precipitated from the filtrate using 78% ethanol. Both fractions were dried, weighed, and quantified. Results were expressed as g/100 g of sample [69].

Fatty acid composition was determined by gas chromatography according to the UNI EN ISO 12966-1:2015 [70], UNI EN ISO 12966-2:2011 [71], and UNI EN ISO 12966-4:2015 [72] standards. Fatty acids were first trans-esterified into their methyl esters using base- or acid-catalyzed methylation, as described in ISO 12966-2:2011 [71]. The esters were then analyzed using a gas chromatograph equipped with a flame ionization detector (FID), following the conditions specified in ISO 12966-4. Identification and quantification of fatty acids were based on retention times and comparison with known standards. The results were expressed as relative percentages of total identified fatty acids [72].

Total starch content was determined enzymatically according to AOAC Official Method 996.11 [69]. The procedure involves gelatinization of starch in the sample with heat and α-amylase, followed by enzymatic hydrolysis with amyloglucosidase to glucose. The released glucose was quantified colorimetric using a glucose oxidase-peroxidase reagent. Results were expressed as total starch (g/100 g of sample).

Phytic acid content was determined using an anion exchange method in accordance with AOAC Official Method 986.11/88. The procedure involves extraction of phytic acid with hydrochloric acid, followed by separation using an anion exchange resin column. Phytic acid was eluted, complexed with Fe^3+^, and quantified spectrophotometrically. Results were expressed as phytic acid (g/100 g of sample) [69].

### 4.4. Extraction and HPLC-DAD Analysis of Phenolic Acids

Free (F), soluble-conjugated (SC) and insoluble-bound (IB) phenolic acid composition of the investigated MBPs was analyzed using the method of Mattila [73] with some modifications [74].

Briefly, each sample was ground to a fine powder using an IKA Tube Mill Control (IKA-Werke GmbH & Co., Staufen, Germany) to achieve a sample homogeneity. Using the quartering method, the extraction was performed on 0.5 g of sample. Hence, 7 mL of a mixture of methanol and 10% acetic acid (85:15) was added to the sample and homogenized for 1 min at 6000 rpm using an IKA T18 digital Ultra Turrax (IKA-Werke GmbH & Co., Staufen, Germany). The homogenized sample was left at room temperature (RT) for 15 min, then sonicated for 30 min at 37 kHz (Elmasonic S 60 H, Elma Schmidbauer GmbH, Singen, Germany), and finally adjusted to a final volume of 10 mL using distilled water. After mixing, 2 mL of the extract was filtered using a 0.45 µm syringe filter. The filtered extracts were used for the HPLC-DAD (Jasco Corporation, Tokyo, Japan) analysis of the free form of phenolic acids. After that, 10 mL of distilled water and 5 mL of NaOH 10 M were added to the crude extract, and sample was stirred for 16 h at RT using a magnetic stirrer M2-A (Argo Lab, Carpi Modena, Italy). The extracted solution was adjusted to pH 2 and the soluble-conjugated and insoluble bound forms of phenolic acids were extracted three times using an equal volume of a mixture Diethyl Ether:Ethyl Acetate (1:1). The three obtained organic layers were combined, and the organic solvent was removed using a Heidolph Hei-VAP Value rotary evaporator (Heidolph Instruments Germany, Schwabach, Germany). The dry extract was dissolved in 1.5 mL of the mixture methanol: acetic acid solution and filtered using a 0.22 µm syringe filter. The extracts were used for the HPLC-DAD analysis of the soluble-conjugated and insoluble bound forms of phenolic acids.

A volume of 20 µL for each sample was injected into a Jasco system composed of a LC-4000 combined with an MD-4015 for HPLC analysis (Jasco Corporation, Tokyo, Japan). The pump, the autosampler, the oven and the diode array detector were managed by the ChromNAV software (Chromatography Data System, Ver. 2, Jasco (Tokyo, Japan). To detect and quantify *p*-coumaric and ferulic acids, two wavelengths were used, 280 nm and 329 nm, respectively. The analyte separation was performed at 30 °C using a 100 × 4.6 mm, 3.5 µm, Zorbax Eclipse Plus C18 (Agilent, Santa Clara, CA, USA) column and a mobile phase composed by H_3_PO_4_ 50 mM at pH 2.5 (solution A) and acetonitrile (solution B). The following gradient elution, at a flow rate of 0.7 mL/min, was applied: isocratic elution 97% A, 0–6 min; linear gradient from 97% to 85% A, 6–17 min; linear gradient from 85% to 82.2% A, 17–27 min, linear gradient from 82.2% to 50% A, 27–40 min. After each run a reconditioning time of 7 min was performed. The analytes were quantified using a linear regression model obtained using different standard concentrations of each analyte.

### 4.5. α-Amylase Quantitation Assay

The quantification of α-amylase activity was carried out using the Ceralpha Kit (Megazyme, Bray, Ireland). The analysis was performed as reported in [62]. First, enzyme was extracted by adding 9 mL of extraction buffer (1 M sodium malate; 1 M sodium chloride; 40 mM calcium chloride; 0.1% sodium azide pH 5.4) to 1.5 g of each sample. Then, the samples were incubated at 40 °C for 20 min and centrifuged at 1000× *g* for 10 min. The test was carried out in a 48-well microplate where, after incubating for 5 min with 30 μL of amylase HR reagent, obtained by dissolving 54.5 mg of BPNPG7 (non-reducing-end blocked p-nitrophenylmaltoheptaoside) and thermostable α-glucosidase (125 U at pH 6.0) per well; and 30 μL of the extracted or control samples were added to the corresponding well. After an incubation of 40 min, 450 μL of “stopping buffer” 20% [*w*/*v*] trisodium phosphate pH ~ 11 solution) was added. The absorbance of samples was read at wavelength of 400 nm using a Tecan Infinite 200 PRO (Tecan, Männedorf, Switzerland) spectrophotometer. The assay was repeated three times for each sample. The obtained data were measured in units of enzyme activity required to release one µmol of para-nitrophenol from BPNPG7 in one minute in the presence of an excess of the thermostable α-glucosidase, which is defined as Ceralpha Unit (CU).

### 4.6. ELISA Quantification of 7S Globulins

Proteins were extracted in triplicate, and all samples were extracted using sodium acetate buffer (50 mM) containing 2 mM L-cysteine. In particular, 0.125 g of each sample was homogenized in 0.5 mL of extraction buffer, using a T25 Ultra-Turrax (IKA-Werke GmbH & Co., Staufen, Germany) for at least 1 min. Each sample was subsequently subjected to centrifugation at 4500× *g* for 30 min at 4 °C. The supernatant of each sample was subsequently analyzed using the following protocol. 7S globulins were determined by two affinity-purified polyclonal antibodies produced in rabbits by GenScript (Rijswijk, The Netherlands), using two antigenic determinants according to literature [75]. Antibodies were tested in Western blotting before performing ELISA tests. ELISA assay was carried out in triplicate as described previously [76] with minor modifications. Briefly, a 96-well plate was incubated overnight at 4 °C with extracted proteins in 50 mM Tris-HCl pH 8 (100 µL/well). Wells were washed twice with PBS (Phosphate buffered saline) buffer then incubated 45 min at RT with 200 µL/well of 5% defatted dry milk dissolved in PBS. Wells were washed twice, 100 µL of diluted serum (Polyclonal Antibody Rabbit-Anti-7S_2, Genescript, The Netherlands) were added to each well (final dilution, in PBS 1:750) and incubated at room temperature for 2 h. Then, wells were washed three times with 0.05% Tween 80 in PBS and incubated for 90 min at RT with 100 µL of 1:3000 diluted HRP-conjugate secondary antibody to each well (Anti-Rabbit IgG Peroxidase antibody produced in goat, Sigma Aldrich, Milan, Italy). After washing as above, plates were washed once with 100 mM sodium acetate buffer pH 6. Color development was performed at room temperature (RT) by adding 100 µL of developing solution (0.3 mM 3,30,5,5-Tetramethylbenzidine, 0.03% (*v*/*v*) hydrogen peroxide (H_2_O_2_) in 100 mM CH_3_COONa pH 6.0). Reaction was stopped by adding 25 µL of 5N H_2_SO_4_. The absorbance value was read at wavelength of 450 nm using a Tecan Infinite 200 PRO (Tecan, Männedorf, Switzerland) spectrophotometer.

### 4.7. Statistical Analysis

The values of each considered parameter, in triplicate, were compared among the MBPs, both within and among commercial mixtures. All the analyses were performed using R software ver. 4.4.0 [77]. Data distribution was tested for normality with the Shapiro–Wilk test, and the homogeneity of variances was assessed by Bartlett’s test. Normally distributed data were processed using one-way ANOVA using “aov” function, followed by Tukey’s Honestly Difference (HSD) post hoc test using “TukeyHSD” function, both present in “stats” package. Non-normally distributed data were compared using the Kuskal–Wallis test followed by Dunn’s post hoc test. In all the analyses, the null hypothesis was considered rejected for *p*-values < 0.05, and the significance level of the comparisons was converted into a compact letter display (CLD) using “multcompView” package. Graphs were produced in the RStudio (ver. 2024.04.0+735) environment using the libraries “ggplot2”, “ggrepel”, “ggtext”, and “tidyverse”.

## 5. Conclusions

The presence of valuable compounds in agri-food MBPs, which could have a positive impact on human health, makes their reuse an important opportunity not only for the environment, but also for human wellbeing [11,78]. To date, most of the literature has focused on the MBPs as a whole, under the generic term of “bran”. However, depending on the milling procedures, wheat by-products can be further separated into fractions with different nutritional and technological properties. We hereby demonstrated that these fractions show important differences according to the commercial cultivars and mixture of grains considered, and such differences could and should be exploited by the food and food supplements industries. Thus, our results represent a steppingstone in the exploration of the potential of wheat by-products and their possible valorization not only in animal farming, but also in human nutrition, paving the way for further research on their implementation in wheat products and/or fortified foods.

However, in order to fully exploit these MBPs, more efforts are needed in developing technologies and solutions to incorporate these matrices into food to meet the organoleptic characteristics required by the consumers [79,80,81,82,83]. Moreover, the antioxidant potential of the phenolic acids could be more efficiently exploited by developing green extractions capable of breaking down their ester bound to the cell wall fibers, thus providing both bio-available phenolic acids and insoluble fibers with unaltered prebiotic potential [38,84]. To achieve this in an environmentally friendly and sustainable way, enzymatic hydrolysis is one of the most attractive techniques as it is highly selective, does not alter the initial organoleptic properties of the bran, and is easy to scale up [85,86].

## Figures and Tables

**Figure 1 ijms-26-05830-f001:**
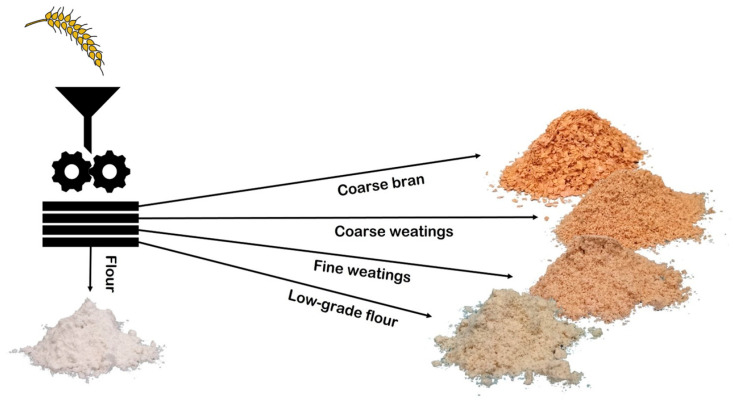
Representation of the four different by-products deriving from the roller milling process. The purpose of this milling process is the production of flour which represents the final and desirable product of this process.

**Figure 2 ijms-26-05830-f002:**
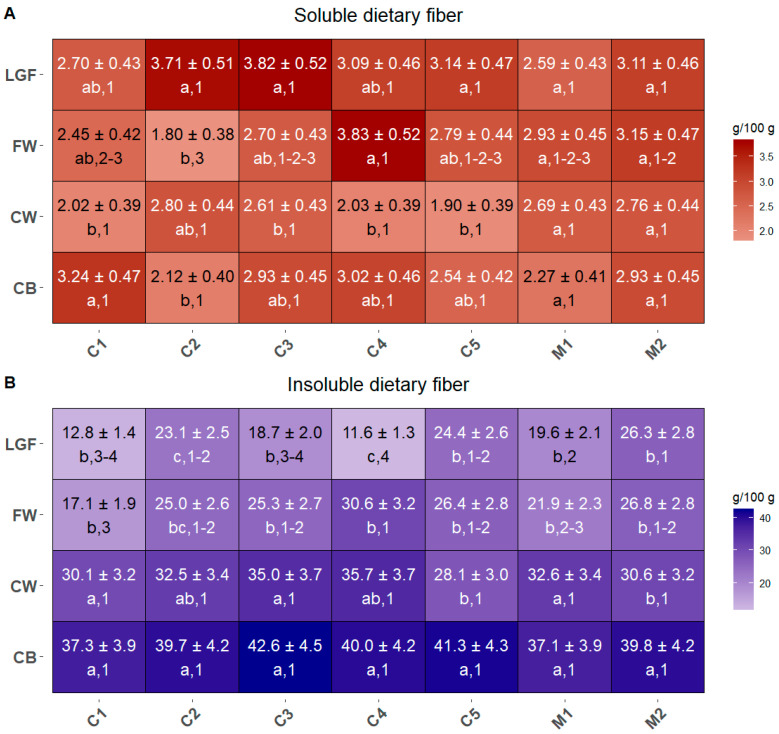
Concentration of soluble (**A**) and insoluble (**B**) dietary fibers contained in the MBPs of all commercial cultivar and mixtures. Data are expressed as g/100 g DW. Different letters (shown in the line below the values) indicate significant differences among MBPs within each cultivar/mixture, while different numbers (also shown below the values) indicate significant differences between cultivars/mixtures for each MBP. Statistical significance was assessed using one-way ANOVA followed by Tukey’s HSD post hoc test (*p* < 0.05).

**Figure 3 ijms-26-05830-f003:**
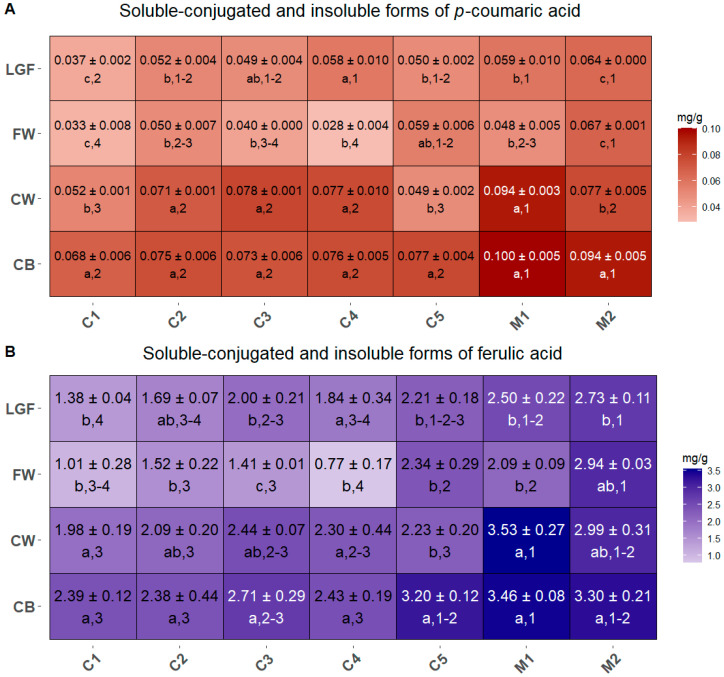
Concentration of soluble-conjugated and insoluble forms of *p*-coumaric acid (**A**) and ferulic acid (**B**) contained in the MBP of each commercial mixture. Data are expressed as mg/g dry weight. Different letters (shown in the line below the values) indicate significant differences among MBPs within each cultivar/mixture, while different numbers (also shown below the values) indicate significant differences between cultivars/mixtures for each MBP. Statistical significance was assessed using one-way ANOVA followed by Tukey’s HSD post hoc test (*p* < 0.05).

**Figure 4 ijms-26-05830-f004:**
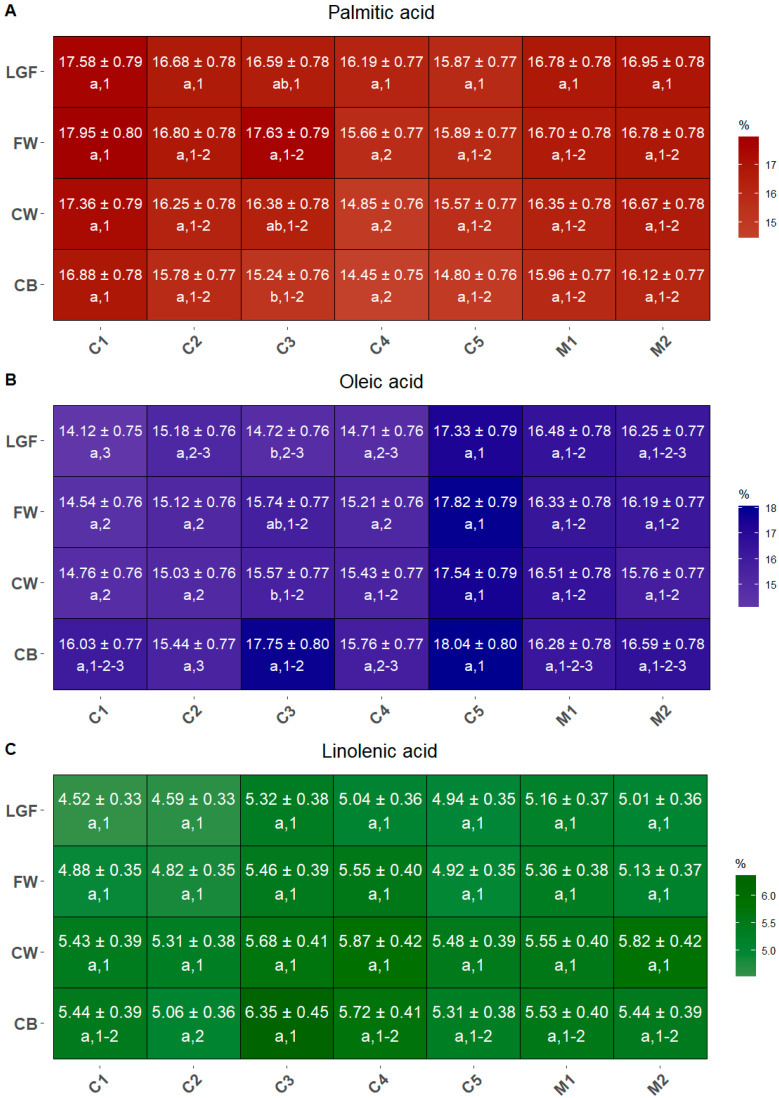
The concentration of palmitic acid (**A**), oleic acid (**B**), and linolenic acid (**C**) contained in the MBPs of all commercial mixtures, expressed as a percentage over the total lipidic content. Different letters (shown in the line below the values) indicate significant differences among MBPs within each cultivar/mixture, while different numbers (also shown below the values) indicate significant differences between cultivars/mixtures for each MBP. Statistical significance was assessed using one-way ANOVA followed by Tukey’s HSD post hoc test (*p* < 0.05).

**Figure 5 ijms-26-05830-f005:**
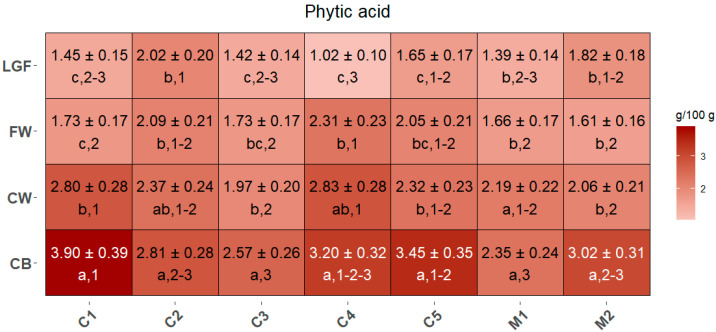
Concentration of phytic acid contained in MBPs of each commercial mixture, expressed in g/100 g of sample dry weight. Different letters (shown in the line below the values) indicate significant differences among MBPs within each cultivar/mixture, while different numbers (also shown below the values) indicate significant differences between cultivars/mixtures for each MBP. Statistical significance was assessed using one-way ANOVA followed by Tukey’s HSD post hoc test (*p* < 0.05).

**Figure 6 ijms-26-05830-f006:**
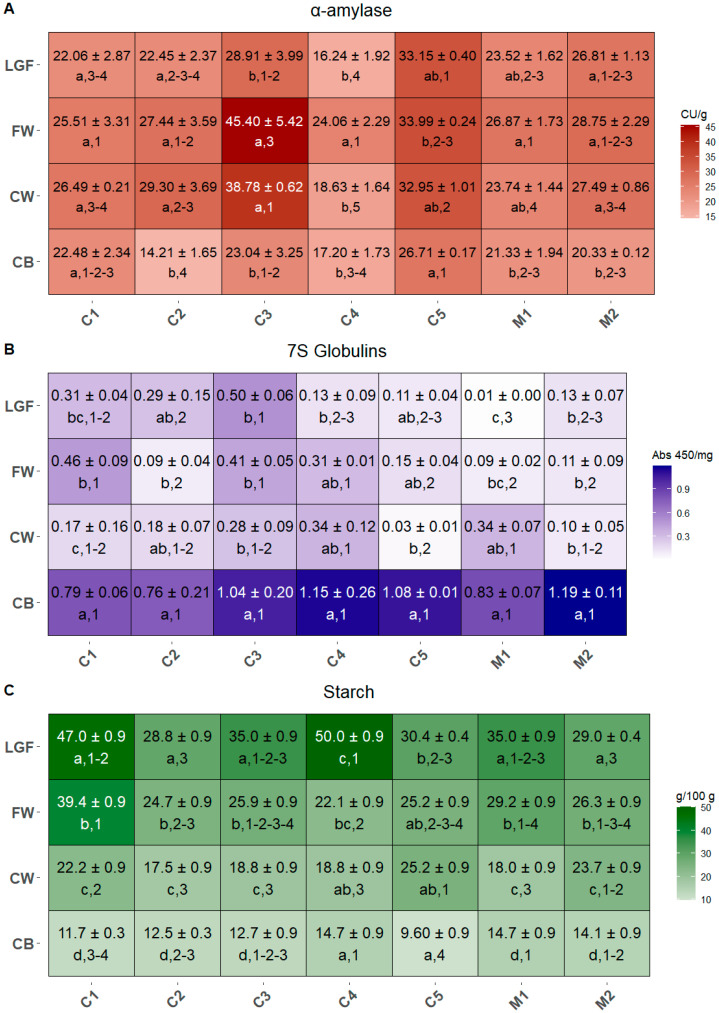
Concentration of α-amylase (**A**), 7S globulins (**B**), and starch (**C**) contained in the MBPs of all commercial mixtures. Data are expressed as CU/g, Abs450/mg and g/100 g dry weight, respectively. Different letters (shown in the line below the values) indicate significant differences among MBPs within each cultivar/mixture, while different numbers (also shown below the values) indicate significant differences between cultivars/mixtures for each MBP. Statistical significance was assessed using one-way ANOVA followed by Tukey’s HSD post hoc test (*p* < 0.05).

**Table 1 ijms-26-05830-t001:** Names, composition, and origin of the cultivar and commercial mixtures analyzed.

Sample Code	Commercial Cultivar or Mixture	Commercial Cultivar/Mixture Composition	Origin
**C1**	Axioma	Axioma	Germany
**C2**	Capo Austriaco	Capo Austriaco	Austria
**C3**	Rumeno	Rumeno	Romania
**C4**	Soisson	Soisson	Italy
**C5**	Manitoba	Manitoba	Canada
**M1**	Misto Rosso Baby Food	Adelante, Africa, Agadir, Akaman, Alcione, Andalusia, Amarok, Aubusson, Bandera, Bilancia, Califasur, Centauro, Guadalupe, Hyxo, Mirroir, Mogal, Moisson, Palesio, PR 22, Soisson, Solehio, Sorial, Zanzibar, Nogall, Ilaria, Basmati	Italy
**M2**	Misto Ungherese	Several commercial mixtures, with a prevalence of Euclide	Hungary

## Data Availability

Data is contained within the article.

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
