# Peer review of "Biochemical Analysis of Wheat Milling By-Products for Their Valorization as Potential Food Ingredients"

_ijms, 2025, doi:10.3390/ijms26125830_

Round 1

Reviewer 1 Report

Comments and Suggestions for Authors

Manuscript Number: ijms-3647025

Manuscript title is “Biochemical Analysis of Wheat Milling by-Products for their Valorization as Potential Food Ingredients”

This study aimed to analyze the macronutrient and bioactive compound profiles of coarse bran, coarse weatings, fine weatings, and low-grade flour across five commercial wheat cultivars and two commercial wheat mixtures, assessing their potential for human nutrition.

 Here are some comments on the manuscript:

  • Revise the abstract, mentioning the most important results obtained.
  • Lines 269-271: While cohort studies have demonstrated that insoluble cereal fiber and whole-grain products might decrease the risk of some non-communicable diseases (e.g., type 2 diabetes, cancer, and cardiovascular diseases), please add appropriate references.
  • Lines 275-284: Indeed, in our analyses, the concentration of insoluble fibers in MBPs resulted on average approximately 11 times higher than that of soluble ones,………add the results, or refer to them in the relevant Figures.
  • Lines 285-286: As concerns soluble fibers, they did not show a univocal trend in their distribution among different MBPs or commercial cultivars and mixtures (Fig. 2A). This sentence needs clarification and scientific explanation.
  • Lines 316- 318: It is now known that its antinutritional effect is only relevant in subjects with a low daily intake of minerals. Add appropriate references, and what do you mean by low daily intake?
  • Line 342-343: ……and consist mainly composed of two essential fatty acids, i.e., linoleic and linolenic, and of oleic acid. review this paragraph.
  • Line 355: Replace ambiguous with another suitable word.
  • The manuscript contained a large number of references.

Author Response

Manuscript title is “Biochemical Analysis of Wheat Milling by-Products for their Valorization as Potential Food Ingredients” This study aimed to analyze the macronutrient and bioactive compound profiles of coarse bran, coarse weatings, fine weatings, and low-grade flour across five commercial wheat cultivars and two commercial wheat mixtures, assessing their potential for human nutrition.

Comment: Revise the abstract, mentioning the most important results obtained.

Response: Thank you very much for your valuable feedback. We have added the most important results to the abstract and partially revised it to better highlight these findings.

Comment: Lines 269-271: While cohort studies have demonstrated that insoluble cereal fiber and whole-grain products might decrease the risk of some non-communicable diseases (e.g., type 2 diabetes, cancer, and cardiovascular diseases), please add appropriate references.

Response: Thank you for your valuable comment. We have now added two appropriate references to support the statement regarding the potential protective effects of insoluble cereal fiber and whole-grain products.

Comment: Lines 275-284: Indeed, in our analyses, the concentration of insoluble fibers in MBPs resulted on average approximately 11 times higher than that of soluble ones,………add the results, or refer to them in the relevant Figures.

Response: Thank you very much for your helpful suggestion. We have now included the reference to the relevant figure where the results on the concentration of insoluble and soluble fibers in MBPs are shown.

Comment: Lines 285-286: As concerns soluble fibers, they did not show a univocal trend in their distribution among different MBPs or commercial cultivars and mixtures (Fig. 2A). This sentence needs clarification and scientific explanation.

Response: Thank you for your comment. We have revised the text accordingly, and we hope it is now clearer and more scientifically accurate.

Comment: Lines 316- 318: It is now known that its antinutritional effect is only relevant in subjects with a low daily intake of minerals. Add appropriate references, and what do you mean by low daily intake?

Response: We thank the reviewer for this insightful comment. In response, we have revised the sentence for greater clarity and added appropriate references to support the statement. Specifically, we clarified that the antinutritional effect of phytic acid is primarily significant in individuals whose mineral intake falls below the recommended dietary allowances.

Comment: Line 342-343: ……and consist mainly composed of two essential fatty acids, i.e., linoleic and linolenic, and of oleic acid. review this paragraph.

Response: Thank you very much for your feedback. We have revised the paragraph as suggested and hope that it is now clearer and easier to understand.

Comment: Line 355: Replace ambiguous with another suitable word.

Response: Thanks for the comment. We replace “ambiguous” with more appropriate “unclear”.

Comment: The manuscript contained a large number of references.

Response: Thank you for your comment. We appreciate your feedback regarding the number of references. However, we believe that all the references included are appropriate to properly frame and support our work. Moreover, we would like to note that Reviewer 2 has requested the addition of further specific references, which we have taken into consideration to ensure a comprehensive and well-contextualized manuscript.

Reviewer 2 Report

Comments and Suggestions for Authors

To the authors, I commend you for developing a well-structured and well-written paper. Below are some of my comments so that you can further improve your work.

General Comments:

  • Although minor, kindly double-check the consistency of the decimal places used throughout the manuscript.
  • Ensure that all chemical formulas are formatted correctly.
  • Overall, cited references are appropriate and mostly up to date.

Title:

  • The title is catchy, and the words were carefully chosen to reflect what the research was all about

Abstract:

  • Abstract is excellent in my opinion.

Keywords:

  • Add wheat if the maximum number of keywords has not yet been reached.

Introduction:

  • Overall, the introduction is comprehensive, and the topic was defined accordingly. Key studies were cited, and the research gap was thoroughly identified.
  • Aims and objectives are pretty vague. Perhaps, the author can specify the tests or analyses that were performed.
  • Also, mentioning economic potential can be misleading in the aims and objectives part, as this was not directly measured or analyzed. Please revise.

Methodology:

  • Please create a separate section that details all the chemicals and reagents used since this paper is heavily geared toward analytical chemistry.
  • Kindly describe how each MBP was prepared/extracted before each analysis.
  • Section 4.2. Quantification of fibers, fatty acids, starch, and phytic acid. These methodologies are considered the central theme of the paper. Therefore, the authors must disclose and detail each method to reproduce it in their laboratories.
  • As much as possible, please cite the paper/s from which the methodologies were derived.
  • In all quantification methodologies, ensure that the guidelines used for the analysis are mentioned.
  • Apart from the three points, the rest of the methodologies are very detailed, and replication and statistical analysis were detailed accordingly.
  •  

Results:

  • All the letters and numbers that connotate significance should be defined.
  • Please increase the size of the figures/graphs as much as possible. The font size should also be enlarged because the current version has a tiny font.
  • As much as possible, include only one graph per figure.
  • The result overall was well written.

Discussion:

  • Discussion is comprehensive, results were interpreted substantially, and comparison of research findings with other literature was very evident.

Author Response

To the authors, I commend you for developing a well-structured and well-written paper. Below are some of my comments so that you can further improve your work.

Comment: Although minor, kindly double-check the consistency of the decimal places used throughout the manuscript.

Response: Thank you for your observation. We have carefully reviewed the manuscript and checked the consistency of all decimal places, making adjustments where necessary.

Comment: Ensure that all chemical formulas are formatted correctly.

Response: Thank you for constructive comment. We have reviewed all the chemical formulas and corrected them where necessary to ensure proper formatting and accuracy.

Comment: Overall, cited references are appropriate and mostly up to date.

Response: Thank you for your positive feedback. We are glad that the references were found to be appropriate and mostly up to date.

Comment: The title is catchy, and the words were carefully chosen to reflect what the research was all about

Response: Thank you very much for your kind comment. We appreciate your positive feedback on the title. We put careful thought into selecting the words to ensure they accurately conveyed the essence of the research.

Comment: Abstract is excellent in my opinion.

Response: Thank you for your generous feedback on the abstract.

Comment: Add wheat if the maximum number of keywords has not yet been reached.

Response: Thank you for this helpful suggestion. We considered including "wheat" among the keywords, but since we had reached the maximum number allowed, we opted to include "wheat" in the title instead, while keeping Triticum aestivum in the keywords. This choice was made to enhance the paper’s discoverability through search engines after publication by incorporating both the common and scientific names across the title and keywords.

Comment: Overall, the introduction is comprehensive, and the topic was defined accordingly. Key studies were cited, and the research gap was thoroughly identified.

Aims and objectives are pretty vague. Perhaps, the author can specify the tests or analyses that were performed.

Response: Thank you very much for your valuable feedback. We appreciate your suggestion regarding the clarity of our aims and objectives. In response, we have revised the relevant section to specify the tests and analyses performed in the study, ensuring that our objectives are now clearly defined and more aligned with the methods employed.

Comment: Also, mentioning economic potential can be misleading in the aims and objectives part, as this was not directly measured or analysed. Please revise.

Response: Thank you for your valuable comment. We appreciate your observation regarding the reference to economic potential. We have revised the text accordingly and included a clarification to avoid any misleading implications, ensuring that the aims and objectives accurately reflect the scope of our analysis.

Comment: Please create a separate section that details all the chemicals and reagents used since this paper is heavily geared toward analytical chemistry.

Response: Thank you for your comment. As requested, we have indicated in a separate section where all chemical reagents were purchased, unless otherwise specified.

Comment: Kindly describe how each MBP was prepared/extracted before each analysis.

Response: Thank you for your comment. More details regarding the milling process and sieve sizes have been added in Material and Methods sections. Specifically, the process follows the standard industrial wheat milling scheme. This involves using roller mills, plansifters, and purifiers in that order to produce particles of an increasingly fine size, from coarse bran to low-grade flour. MBPs are collected at specific stages of the process according to particle size. All resulting products are stored in by-product silos under the same conditions. In order to obtain “reground” samples, an additional step must be considered compared to the classic milling process. In particular, the larger fractions (coarse bran and coarse weatings) undergo a further regrinding step through a dedicated rolling mill, followed by sieving operations to separate them into the various final products, which are determined by particle size.

Comment: Section 4.2. Quantification of fibers, fatty acids, starch, and phytic acid. These methodologies are considered the central theme of the paper. Therefore, the authors must disclose and detail each method to reproduce it in their laboratories.

Response: Thank you for your valuable comment. In response to your suggestion, we have added as much methodological detail as possible for each analysis to ensure clarity and reproducibility. However, since the procedures employed are based on internationally recognized standardized methods (e.g., AOAC and ISO standards), we followed the official protocols as prescribed. These methods are widely validated and publicly documented, which unfortunately limits our ability to reproduce every procedural detail within the manuscript. Nonetheless, we have provided the relevant references so that readers can access the full protocols if needed.

Comment: As much as possible, please cite the paper/s from which the methodologies were derived. In all quantification methodologies, ensure that the guidelines used for the analysis are mentioned.

Response: Thank you for your comment. We have ensured that for each quantification methodology, where applicable, we have included a bibliographic reference to the relevant guidelines or sources used in the analysis.

Comment: Apart from the three points, the rest of the methodologies are very detailed, and replication and statistical analysis were detailed accordingly.

Response: Thank for your positive comment.

Comment: All the letters and numbers that connotate significance should be defined.

Response: Thank you for this observation. To improve clarity, we have now explicitly stated in the caption that differences were considered significant at p < 0.05, as also described in the Methods section. The updated caption now reads: “Different letters (shown in the line below the values) indicate significant differences among MBPs within each cultivar/mixture, while different numbers (also shown below the values) indicate significant differences between cultivars/mixtures for each MBP. Statistical significance was assessed using one-way ANOVA followed by Tukey’s HSD post hoc test (p < 0.05).” We hope this revision provides a clearer explanation of the statistical significance used in the figures.

Comment: Please increase the size of the figures/graphs as much as possible. The font size should also be enlarged because the current version has a tiny font.

Response: We thank you for the suggestion. The font sizes of the graphs, including axes, titles, and value labels, have been increased. In particular, the size of the value labels has been enlarged as much as possible, considering the space available within the boxes where the values are displayed.
Additionally, to further improve readability, the font colour of the value labels now switches between black and white depending on the background colour, enhancing contrast and making the values easier to read.

Comment: As much as possible, include only one graph per figure.

Response: Thank you very much for your valuable comment. While we understand the suggestion to include only one graph per figure, we believe that, in this case, presenting two or more graphs within a single figure allows for a more coherent and effective comparison of the data. Therefore, we would prefer to retain the current figure layout, as we feel it enhances visual clarity and strengthens the overall presentation.

Comment: The result overall was well written.

Response: Thank for your positive comment.

Comment: Discussion is comprehensive, results were interpreted substantially, and comparison of research findings with other literature was very evident.

Response: Thank you for your positive feedback. We appreciate your recognition of the depth of our discussion, the interpretation of the results, and the comparison with existing literature. Your comments are greatly encouraging.

Reviewer 3 Report

Comments and Suggestions for Authors

Major Comment:
This manuscript focuses solely on analyzing the dietary fibers, free and bound phenolics, phytic acid, fatty acids, and aleurone layer markers in wheat milling by-products. However, it does not explore the potential applications of these by-products, such as their bioactivities experiments or environmental aspects like carbon footprint. As a result, the novelty and research depth of the study are somewhat limited. Therefore, I do not recommend the manuscript for publication in its current form.

Minor Comment:

The authors should clarify the purpose and significance of analyzing aleurone layer markers, particularly in terms of their scientific or practical (industrial) relevance.

Author Response

REV3: This manuscript focuses solely on analysing the dietary fibers, free and bound phenolics, phytic acid, fatty acids, and aleurone layer markers in wheat milling by-products.

Comment: However, it does not explore the potential applications of these by-products, such as their bioactivities experiments or environmental aspects like carbon footprint. As a result, the novelty and research depth of the study are somewhat limited.

Response: Thank you for your insightful comment. We would like to clarify that the primary aim of our manuscript is to provide a detailed and comparative biochemical analysis of the nutritional and bioactive components present in wheat milling by-products, with a focus on the distribution among dietary fiber, free and bound phenolics, phytic acid, fatty acids, and aleurone markers. This type of characterization is a fundamental and preliminary step for any future functional enhancement or industrial application, including nutraceutical, food or environmental applications. This study suggests a possible reuse of these matrices and therefore their valorisation.

We recognize that your suggestions regarding the assessment of bioactivities and environmental factors, such as carbon footprint, are extremely relevant. However, we believe that these aspects deserve dedicated and specific insights that are beyond the scope of this manuscript. Anyway, our results can provide a solid basis for subsequent studies geared toward functional enhancement and sustainability.

Comment: Therefore, I do not recommend the manuscript for publication in its current form.

Response: We thank you for the time and effort you dedicated to reviewing our manuscript. We are naturally disappointed that, in its original form, the work was not considered suitable for publication. However, we have carefully revised the manuscript in response to the reviewers’ comments, and in particular have made substantial changes based on the suggestions provided by the other two reviewers. We hope that, in light of these revisions, you might reconsider your evaluation of our work.

Comment: The authors should clarify the purpose and significance of analysing aleurone layer markers, particularly in terms of their scientific or practical (industrial) relevance.

Response: Thank you for your thoughtful comment. We have clarified the purpose and significance of analysing aleurone layer markers in the revised text. In particular, we have emphasized their scientific relevance, as well as their practical implications in potential industrial applications related to by-products.

Round 2

Reviewer 1 Report

Comments and Suggestions for Authors

The authors overlooked or ignored some comments and dealt superficially with others. For example:

1- Comments 1, 2 and 4.
 2- Comment 5, a suitable reference should be added to the end of the sentence: "It is now known that its antinutritional effect is only relevant in subjects with a low daily intake of minerals."
3- Comment 7, the replacement of the word requires clarification of the meaning of this sentence.

Author Response

Comments 1: Revise the abstract, mentioning the most important results obtained.

R: We thank the reviewer for the helpful comment. Following the suggestion, we have revised the content of the abstract to better highlight the most important results obtained in the study. We hope the new version is now clearer and more aligned with the reviewer’s expectations.

Comment 2: Lines 269-271: While cohort studies have demonstrated that insoluble cereal fiber and whole-grain products might decrease the risk of some non-communicable diseases (e.g., type 2 diabetes, cancer, and cardiovascular diseases), please add appropriate references.

R: Thank you for your comment and suggestion. In response, we have added appropriate references to support the statement regarding the potential protective role of insoluble cereal fiber and whole-grain products against non-communicable diseases such as type 2 diabetes, cancer, and cardiovascular diseases. The following references have been included in the revised manuscript (Line 304):

  • Fatima, I.; Gamage, I.; De Almeida, R.J.R.; Cabandugama, P.; Kamath, G. Current Understanding of Dietary Fiber and Its Role in Chronic Diseases. Mo Med 2023, 120, 381-388, doi:PMID: 37841565; PMCID: PMC10569388.
  • Partula, V.; Deschasaux, M.; Druesne-Pecollo, N.; Latino-Martel, P.; Desmetz, E.; Chazelas, E.; Kesse-Guyot, E.; Julia, C.; Fezeu, L.K.; Galan, P.; et al. Associations between consumption of dietary fibers and the risk of cardiovascular diseases, cancers, type 2 diabetes, and mortality in the prospective NutriNet-Sante cohort. Am J Clin Nutr 2020, 112, 195-207, doi:10.1093/ajcn/nqaa063.

Comment 4: Lines 285-286: As concerns soluble fibers, they did not show a univocal trend in their distribution among different MBPs or commercial cultivars and mixtures (Fig. 2A). This sentence needs clarification and scientific explanation.

R: Thank you for your insightful comment regarding the description of the results on soluble fibers. In response, we have revised the sentence in the manuscript to improve clarity and adopt a more positive and constructive tone. The revised sentence now reads (Lines 318-320):

“Although soluble fibers did not show a clear pattern across the different MBPs or commercial cultivars and mixtures (Fig. 2A), this variability could represent a particularly promising opportunity for further exploration.”

Comment 5: a suitable reference should be added to the end of the sentence: "It is now known that its antinutritional effect is only relevant in subjects with a low daily intake of minerals."

R: We thank the reviewer for this valuable observation. We apologize for the oversight. Unfortunately, the reference was unintentionally omitted during the text formatting process. We have now correctly inserted the appropriate following citation (Line 351): Silva, E.O.; Bracarense, A.P.F.R.L. Phytic Acid: From Antinutritional to Multiple Protection Factor of Organic Systems. J Food Sci 2016, 81, R1357-R1362, doi:10.1111/1750-3841.13320.

Comment 7: the replacement of the word requires clarification of the meaning of this sentence.

R: We thank the reviewer for pointing out the need for clarification. We have revised the sentence to improve its clarity and hope that the new formulation better conveys the intended meaning. In the revised version, the sentence (Lines 392-395) reports: “Palmitic acid, a saturated fatty acid found in wheat grains, has an ambiguous nutraceutical value. On one hand, as a saturated fat, it may contribute to increased blood cholesterol levels and negatively impact cardiovascular health. On the other hand, it also plays a role in regulating lipid metabolism and inflammatory responses” [53]

Reviewer 3 Report

Comments and Suggestions for Authors

I have already expressed my comments in the previous report and have no further comments.

Author Response

Dear Reviewer, 

we had already responded in the previous revision.

As previously reported, we believe that your requests, while certainly insightful, are not fully applicable to the current scope of this article being the article focused on detailed and comparative biochemical analysis of the nutritional and bioactive components present in wheat milling by-products, with a focus on the distribution among dietary fiber, free and bound phenolics, phytic acid, fatty acids, and aleurone markers. This type of characterization is a fundamental and preliminary step for any future functional enhancement or industrial application, including nutraceutical, food or environmental applications. This study suggests a possible reuse of these matrices and therefore their valorisation.

Nevertheless, we acknowledge the valuable perspectives offered in the review and are already planning the necessary experiments to address these suggestions in future paper.

Round 3

Reviewer 1 Report

Comments and Suggestions for Authors

No additional comments